# "Outcomes of COVID-19 infection in patients with hematological malignancies- A multicenter analysis from Pakistan"

Adeeba Zaki[1,2]*, Salman Muhammad Soomar[1,2], Danish Hasan Khan[3,4], Hasan Shaharyar Sheikh[5,6], Raheel Iftikhar[7], Ayaz Mir[8,9], Zeba Aziz[10,11], Khadija Bano[8,12], Hafsa Naseer[10,13], Qamar un–Nisa Chaudhry[7], Syed Waqas Imam Bokhari[5,14], Munira Shabbir-Moosajee[1,2]

1 Aga Khan University Hospital, Karachi, Pakistan, 2 Department of Oncology, The Aga Khan University Hospital, Karachi, Pakistan, 3 Hangzhou Tigermed, Karachi, Pakistan, 4 Clinical Project Manager (CPM Pakistan), Hangzhou Tigermed, Karachi, Pakistan, 5 Shaukat Khanum Memorial Cancer Hospital and Research Centre, Lahore, Pakistan, 6 Consultant Medical Oncologist, Shaukat Khanum Memorial Cancer Hospital and Research Centre, Lahore, Pakistan, 7 Armed Forces Bone Marrow Transplant Centre, Rawalpindi, Pakistan, 8 Shifa International Hospital, Islamabad, Pakistan, 9 Bone Marrow & Stem Cell Transplant, Shifa International Hospital, Karachi, Pakistan, 10 Hameed Latif Hospital, Lahore, Pakistan, 11 Medical Oncologist, Hammed Latif Hospital, Lahore, Pakistan, 12 Fellow Clinical Hematology Shifa International Hospital, Islamabad, Pakistan, 13 Postgraduate Resident, Hameed Latif Hospital, Lahore, Pakistan, 14 Medical Oncologist, Shaukat Khanum Memorial Cancer Hospital and Research Centre, Lahore, Pakistan

* adeeba.zaki@aku.edu

**Data Availability Statement:** All relevant data are within the paper and its Supporting Information files.

## Abstract

### Purpose

COVID-19 infection resulting from severe acute respiratory syndrome coronavirus 2 (SARS-CoV-2) began to spread across the globe in early 2020. Patients with hematologic malignancies are supposed to have an increased risk of mortality from coronavirus disease of 2019 (COVID-19) infection. From Pakistan, we report the analysis of the outcome and interaction between patient demographics and tumor subtype and COVID-19 infection and hematological malignancy.

### Patients and methods

This multicenter, retrospective study included adult patients with a history of histologically proven hematological malignancies who were tested positive for COVID-19 via PCR presented at the oncology department of 5 tertiary care hospitals in Pakistan from February to August 2020. A patient with any known hematological malignancy who was positive for COVID-19 on RT-PCR, was included in the study. Chi-square test and Cox-regression hazard regression model was applied considering p $\leq$ 0.05 significant.

### Results

A total of 107 patients with hematological malignancies were diagnosed with COVID-19, out of which 82 (76.64%) were alive, and 25 (23.36%) were dead. The significant hematological

**Funding:** The authors received no specific funding for this work.

**Competing interests:** The authors have declared that no competing interests exist.

**Abbreviations:** ASH, American Society for Hematology; CI, Confidence Interval; GP, General Physician; HR, Hazard Ratio; ICU, Intensive Care Unit; NHL, Non-Hodgkin's Lymphoma; RT-PCR, Real Time Polymerase Chain Reaction; UK, United Kingdom.

malignancy was B-cell Lymphoma in dead 4 (16.00%) and alive group 21 (25.61%), respectively. The majority of the patients in both the dead and alive group were on active treatment for hematological malignancy while they came positive for COVID-19 [21 (84.00%) & 48 (58.54%) p 0.020]. All patients in the dead group were admitted to the hospital 25 (100.00%), and among these, 14 (56.00%) were admitted in ICU with a median 11 (6–16.5) number of days. Among those who had contact exposure, the hazard of survival or death in patients with hematological malignancies and COVID-19 positive was 2.18 (CI: 1.90–4.44) times and 3.10 (CI: 2.73–4.60) times in patients with travel history compared to no exposure history (p 0.001).

## Conclusion

Taken together, this data supports the emerging consensus that patients with hematologic malignancies experience significant morbidity and mortality resulting from COVID-19 infection.

## Introduction

The SARS-Cov-2 or COVID-19, known as novel coronavirus, has become a global threat and healthcare concern. Since its outbreak in China at the end of 2019, the pandemic has affected more than 100 million people worldwide [1]. Although the outbreak is likely to have started from a zoonotic spread, it soon became apparent that person-to-person transmission occurs mainly through respiratory droplets and direct contact with a diseased person or indirect contact with fomites in the environment [2, 3]. Many people have mild symptoms. In contrast, others have no symptoms at all but still, actively carry and transmit the virus. However, some do develop severe symptoms such as respiratory failure, cytokine release syndrome, and multi-organ failure [4].

Since COVID-19 began to spread across the globe in early 2020, patients with comorbidities and cancer are more susceptible to marked complications of viral infection. Cancer patients are more prone to increased risk of infections than individuals without cancer because of immunosuppression by the malignancy itself and anticancer treatments, such as chemotherapy or surgery [5, 6] and have a poorer prognosis. Cao et al. have reported that patients with metastatic disease, hematological malignancy, or lung cancer are at particularly high risk of severe complications, requiring intensive care unit (ICU) admission, invasive mechanical ventilation, and even death. Furthermore, active treatment such as surgery and immunotherapy is associated with a significantly increased risk (hazard ratios of 6.22 and 4.82) respectively, for poor outcomes [7], Zhang et al. reported a higher rate of adverse events (53.6%) and mortality (28.6%) in those who had their last anti-tumor treatment within 14 days of the infection (HR = 4.079, 95% CI 1.086–15.322, P = 0.037). We do know that patients with hematological malignancies can have an underlying immune dysfunction and are vulnerable to viral and other infections [9]. Additionally, treatments which include cytotoxic agents, immunomodulators, hematopoietic stem cell transplantation, and chimeric antigen receptor T-cell therapy, are profoundly immunosuppressive. In addition, patients with hematologic malignancies have multiple risk factors of particular concern in the context of COVID-19 infection such as advanced age, underlying or treatment-induced comorbid illnesses like hypertension and

diabetes, and chronic lymphopenia. These factors make this patient population particularly susceptible to an adverse outcome.

Public health measures are universally instituted to control the disease spread and aim to decrease preventable hospital admissions. In addition, it is recommended that patients receiving anticancer treatment should have vigorous screening for COVID-19 [8]. However, cancer care encompasses a diverse array of primary tumor types and stages, affecting all age groups of patients, with different prognosis and outcomes. Therefore, labeling all patients with cancer as susceptible to COVID-19 is probably neither reasonable nor informative.

For patients with hematologic malignancies, the overall risk of morbidity and mortality resulting from COVID-19 infection, as well as how this risk varies as a function of age, disease status, type of malignancy, and cancer therapy is being studied [9]. However, data from low- and middle-income countries is sparse. It is hypothesized that lower rates of testing and contact tracing and dearth of medical facilities adequately equipped to manage complicated covid infections can potentially result in higher morbidity and mortality in this high-risk population.

From Pakistan, we report the analysis of the interaction between patient demographics and tumor subtype and COVID-19 infection and outcomes in patients with hematological malignancy. This is a multicenter analysis from five tertiary care hospitals in Pakistan, all of whom have an established cancer center.

## Materials and methods

We retrospectively collected data on all the patients with a history of histologically proven hematological malignancies that tested positive for COVID-19 by RT-PCR and present at the oncology department of 5 tertiary care hospitals in Pakistan: Aga Khan University Hospital Karachi, Armed Forces Bone Marrow Transplant Centre Rawalpindi, Hameed Latif Hospital Lahore, Shifa International Hospital Islamabad, and Shaukat Khanum Memorial Cancer Hospital Lahore from February to August 2020. Data was collected using a standard Performa developed for this study. Demographic, clinical, treatment, and laboratory data and serial samples for viral RNA detection were extracted from the medical records of 107 patients.

Patient with any known hematological malignancy who was positive for COVID-1f on RT-PCR was included in the study. Patients with a clinical or radiological diagnosis of COVID-19 without a positive RT-PCR test, were not included in this analysis.

### Operational definition

**Leukemia** is cancer of the body's blood-forming tissues, including the bone marrow and the lymphatic system [10].

**Lymphoma** is a cancer of the lymphatic system, which is part of the body's germ-fighting network. The lymphatic system includes the lymph nodes (lymph glands), spleen, thymus gland, and bone marrow [11].

**Multiple myeloma** is a cancer of plasma cells. Normal plasma cells are found in the bone marrow and are an important part of the immune system [12].

**Myelodysplastic syndromes** (MDS) are conditions that can occur when the blood-forming cells in the bone marrow become abnormal. This leads to low numbers of one or more types of blood cells [12].

### Clinical classification of suspected or confirmed COVID-19 patients in Pakistan

**Asymptomatic** SARS CoV-2 infection but with no symptoms. Some asymptomatic patients may be pre-symptomatic if tested early (e.g., as part of contact tracing) [13].

**Non-Severe** Oxygen saturation of 94% or greater and respiratory rate of less than 25 breaths/minute [14].

**Severe** Oxygen saturation < 94% and respiratory rate greater than 25 breaths/minute.

Oxygen saturation is maintained by nasal cannulation or simple facemask, no need for non-invasive or invasive mechanical ventilation [14].

**Critical** Respiratory compromise severe enough requiring invasive and non-invasive mechanical ventilation [13].

## Variables

Age, Sex, Hospital, Co-morbidities, Hematological malignancy, treatment, chemotherapy, COVID-19 exposure, presenting complaint, hospital admission, COVID-19 status at death or discharge were the covariates.

## Outcome

**Mortality**—the number of deaths in patients with hematological malignancy diagnosed with COVID-19 infection.

## Statistical analysis

The data was entered and analyzed by using STATA version 16. Calculated medians for all continuous skewed variables. The normality of data was checked through density plots—frequencies with percentages for categorical variables. A Chi-square test was performed to check the association between the covariates and mortality. For survival analysis, a multivariable Cox-regression hazard regression model was applied, and the Kaplan Meir survival curve was used to check the overall survival. P-value ≤0.05 was considered significant.

## Ethical approval

Ethical approval was obtained from the Ethical Review Committee of the Aga Khan University Hospital, Karachi, Pakistan. Data were retrospectively collected through chart review, so there was no need for the informed written consent.

## Results

### Baseline characteristics

A total of 107 patients with hematological malignancies were diagnosed with COVID-19, out of which 82 (76.64%) were alive, and 25 (23.36%) were dead. Among the dead, 12 (48.00%) of the participants were greater than 50 years of age (p 0.027). Most of the participants were males in both the dead and alive group [18 (72.00%) & 54 (68.85%)]. Hypertension was the common comorbidity in both groups [8 (32.00%) & 8 (9.76%) p 0.021]. The significant hematological malignancy was B-cell Lymphoma in dead 4 (16.00%) and alive group 21 (25.61%), respectively (Table 1). The majority of the patients in both the dead and alive group were on active treatment for hematological malignancy while they came positive for COVID-19 [21 (84.00%) & 48 (58.54%) p 0.020]. IV chemotherapy was the common treatment 15 (71.43%) from the dead group, and 29 (61.70%) in the alive group received IV chemotherapy moreover Idarubicin + cytarabine was the common IV chemo-regimen given to patients in both groups [5 (23.81%) & 5 (12.20%)] (Table 2). The presenting complaint in patients with hematological malignancies who catch COVID-19 among the dead group were fever (52.00%) followed by respiratory symptoms 11 (44.00%). However, in the alive group, 18 (21.95%) were asymptomatic, 22 (26.83) presented with fever, 33 (40.24%) presented with respiratory symptoms (p

**Table 1. Baseline and clinical characteristics of patients with hematological malignancies and COVID-19 stratified on the outcome (n = 107).**

| Characteristic | Outcome n (%) | | P-value |
|---|---|---|---|
| | Dead [25 (23.36%)] | Alive [82 (76.64%)] | |
| Age (years) | | | 0.027 |
| 10–30 | 7 (28.00) | 34 (41.46) | |
| 31–50 | 6 (24.00) | 31 (37.80) | |
| >50 | 12 (48.00) | 17 (20.73) | |
| Sex | | | 0.566 |
| Male | 18 (72.00) | 54 (68.85) | |
| Female | 7 (28.00) | 28 (34.15) | |
| Hospital | | | 0.012 |
| AKUH | 8 (32.00) | 9 (10.98) | |
| SKMCH | 6 (24.00) | 48 (58.54) | |
| Shifa Hospital | 6 (24.00) | 16 (19.51) | |
| AFBMTC | 5 (20.00) | 7 (8.54) | |
| HLH | 0 (0.00) | 2 (2.44) | |
| Co-morbidities | | | 0.021 |
| None | 8 (32.00) | 50 (60.98) | |
| HTN | 8 (32.00) | 8 (9.76) | |
| Diabetes | 2 (8.00) | 7 (8.54) | |
| Other | 7 (28.00) | 17 (20.73) | |
| Malignancy | | | 0.222 |
| ALL | 3 (12.00) | 11 (13.41) | |
| AML | 7 (28.00) | 6 (7.32) | |
| CML | 1 (4.00) | 4 (4.88) | |
| CLL | 2 (8.00) | 5 (6.10) | |
| Hodgkins | 2 (8.00) | 17 (20.73) | |
| B- cell lymphoma | 4 (16.00) | 21 (25.61) | |
| T- cell lymphoma | 1 (4.00) | 3 (3.66) | |
| Others | 5 (20.00) | 15 (18.29) | |

0.029). There was a contact exposure in 3 (12.00%) and travel history in 8 (32.00%) of patients in the dead group while 13 (15.85%) had contact exposure and 9 (10.98%) had travel history among the alive group (p 0.042). All patients in the dead group were admitted in the hospital 25 (100.00%) and among these, 14 (56.00%) were admitted in ICU with a median 11 (6–16.5) number of days while 24 (29.27%) alive patients were admitted to hospital and only 4 (4.88%) admitted in ICU (p <0.001). At discharge or death, 4 (16.00%) dead patients were COVID positive, and 30 (36.59) alive patients were positive (p 0.008) (Table 3).

## Overall survival and multivariable regression

The median overall survival among patients with hematological malignancies and COVID-19 positive was 33 (IQR 15–60) days (Fig 1). Upon applying multivariable Cox-proportional hazard regression, age (categorical), malignancy treatment, COVID-19 exposure, and COVID-19 status at death or discharge were significant (Table 4). The hazard ratio of survival or death in patients with hematological malignancies and COVID-19 positive in the age group 10–30 years was 1.81 times (CI: 1.40–2.72) and 1.22 times among age group 31–50 years (CI: 1.14–2.00) compared to age greater than 50 years (p 0.019). The hazard ratio of survival or death in patients with hematological malignancies and COVID-19 positive in patients receiving IV chemotherapy was 1.51

**Table 2. Treatment and management of patients with hematological malignances and COVID-19 stratified on outcome (n = 107).**

| Characteristic | Outcome | | P-value |
|---|---|---|---|
| | Dead [25 (23.36%)] | Alive [82 (76.64%)] | |
| Active treatment | | | 0.020 |
| Yes | 21 (84.00) | 48 (58.54) | |
| No | 4 (16.00) | 34 (41.46) | |
| Treatment | | | 0.797 |
| Chemoimmunotherapy | 5 (23.81) | 13 (27.66) | |
| IV chemotherapy | 15 (71.43) | 29 (61.70) | |
| TKI | 1 (4.76) | 4 (8.51) | |
| Radiation | 0 (0.00) | 1 (2.13) | |
| Chemo regimen | | | 0.551 |
| R-CHOP | 2 (9.52) | 7 (17.07) | |
| R- Bendamustine | 0 (0.00) | 2 (4.88) | |
| R-ICE | 1 (4.76) | 2 (4.88) | |
| R-CEOP | 0 (0.00) | 1 (2.44) | |
| R-EPOCH | 1 (4.76) | 1 (2.44) | |
| Rituximab | 1 (4.76) | 0 (0.00) | |
| CVP | 0 (0.00) | 1 (2.44) | |
| Idarubicin + cytarabine | 5 (23.81) | 5 (12.20) | |
| BFM protocol | 1 (4.76) | 3 (7.32) | |
| HyperCVAD | 4 (19.05) | 5 (12.20) | |
| ABVD | 3 (14.29) | 7 (17.07) | |
| Bendamustine | 0 (0.00) | 2 (4.88) | |
| VRD | 3 (14.29) | 1 (2.33) | |
| Cyclophoshamide +bortezomib | 0 (0.00) | 2 (4.88) | |
| Lenalodomide + carfilzomib | 0 (0.00) | 2 (4.88) | |

(CI: 1.26–2.47) times and 1.46 times who received chemoimmunotherapy (CI: 1.30–2.39) compared to those who received TKI (p 0.031). Among those who had contact exposure, the hazard of survival or death in patients with hematological malignancies and COVID-19 positive was 2.18 (CI: 1.90–4.44) times and 3.10 (CI: 2.73–4.60) times in patients with travel history compared to no exposure history (p 0.001). Moreover, the hazards were higher in patients who did not test for COVID or were positive at discharge or death 2.28 (CI: 1.43–5.63) and 3.41 (CI: 2.12–6.38) compared to those who were negative at discharge or death (p 0.001).

## Discussion

Worldwide, healthcare systems are in an uphill task of dealing with the COVID-19 pandemic, a situation that is going to remain a challenge to all clinicians. The incidence of COVID-19 and outcomes in cancer patients is a topic of great interest. It is evident now that the COVID-19 will be a global health care issue for the foreseeable future, and it is imperative that clinicians understand the complexity of presentations and outcomes of patients with concomitant health issues that make them vulnerable to severe complications. Our study has mainly focused on outcomes of COVID-19 with hematological malignancies in a resource-constrained environment, and it's the first multicenter analysis from Pakistan.

American Society for Hematology (ASH) Research Collaborative COVID-19 Registry analysis states that patients with hematologic malignancies have higher mortality resulting from COVID-19 than in the general population. They reported an overall mortality of 28%,

**Table 3. COVID-19 exposure and status at death/discharge among hematological malignancies patients stratified on outcome (n = 107).**

| Characteristic | Outcome | | P-value |
|---|---|---|---|
| | **Dead [25 (23.36%)]** | **Alive [82 (76.64%)]** | |
| Presenting complaint | | | 0.029 |
| Fever | 13 (52.00) | 22 (26.83) | |
| Respiratory symptoms | 11 (44.00) | 33 (40.24) | |
| GI symptoms | 1 (4.00) | 4 (4.88) | |
| Other | 0 (0.00) | 5 (6.10) | |
| Asymptomatic | 0 (0.00) | 18 (21.95) | |
| COVID-19 exposure | | | 0.042 |
| No | 14 (56.00) | 60 (73.17) | |
| Contact exposure | 3 (12.00) | 13 (15.85) | |
| Travel | 8 (32.00) | 9 (10.98) | |
| Hospital admission | | | <0.001 |
| No | 0 (0.00) | 58 (70.73) | |
| Yes | 25 (100.00) | 24 (29.27) | |
| In hospital stay | | | <0.001 |
| No admission | 0 (0.00) | 58 (70.73) | |
| Ward | 5 (20.00) | 16 (19.51) | |
| SCU | 6 (24.00) | 4 (4.88) | |
| ICU | 14 (56.00) | 4 (4.88) | |
| If yes, Number of days | | | 0.003* |
| (Median, IQR) | 11 (6–16.5) | 0 (0–4) | |
| Recovered & discharged | | | <0.001 |
| No | 25 (0.00) | 2 (2.44) | |
| Yes | 0 (0.00) | 35 (42.68) | |
| Home isolation | 0 (0.00) | 45 (54.88) | |
| COVID status at death/discharge | | | 0.008 |
| Not done | 9 (36.00) | 37 (45.12) | |
| Negative | 12 (48.00) | 15 (18.29) | |
| Positive | 4 (16.00) | 30 (36.59) | |

increasing to 42% in patients with moderate to severe infection [15–17]. However, in the general population, it ranges from 9–29.6%. In the white Hispanic group, the mortality was less than 10%, while in Black and Hispanic groups, it was less than 30% in the general population [18]. Recently published study from Italy stated mortality of 37% patients with hematological malignancy and COVID-19, with higher risk amongst those with older age, progressive disease, or severe infection [19, 20]. Another multicenter study, analyzing outcomes in cancer patients from China reported mortality of 20%, with 41% mortality in the 22 patients with hematologic malignancies [21]. UK Myeloma Forum published their results of 75 patients with COVID-19 and multiple myeloma, with mortality of 55% [22, 23]. In our study, the overall mortality mirrors the data of these studies with an overall mortality of 23% and increasing to approximately 51% in the hospitalized patients and 86% in ICU patients. 50% of deaths were seen in patients younger than 50 years. This is most likely due to the fact that approximately 70% of our patients were in that age group. Therefore, this is likely over-represented. It was interesting to note the relatively low numbers of patients of older age admitted to the hospital in our cohort. It could be hypothesized that treatment was deferred for the older patients with multiple co-morbidities in the initial months of the pandemic if they had a relatively stable clinical course. However, it would be interesting to study the outcomes of patients who had their treatment deferred.

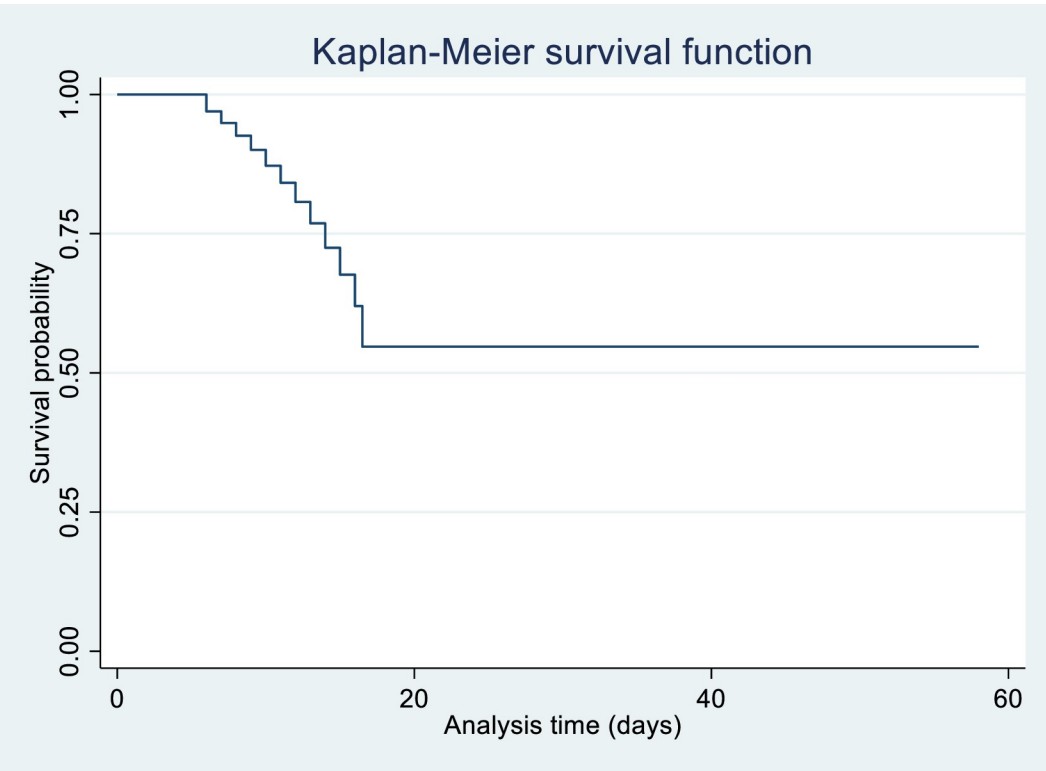

**Fig 1. Graph representing overall survival in patients with hematological malignancies and COVID-19.**

Several other findings from our cohort are noteworthy. We analyzed the demographics of COVID-19 patients with hematological malignancy and explored the effect of cytotoxic chemotherapy and various chemoimmunotherapy and targeted treatments on the trajectory of COVID-19. The incidence of COVID-19 was found to be more frequent in acute leukemias (29%), followed by non-Hodgkin's lymphoma (27%) and Hodgkin's lymphoma (18%) [24]. Furthermore, an increase in mortality has been reported in myeloid malignancies (MDS/AML/MPN) than lymphoid neoplasms (NHL/CLL/ALL/MM/HL) (43% vs. 35%) [25, 26], which is similar to that seen in our study population. The majority of our study patients were on active treatment and reported a mortality rate of 19.6%, in contrast to 10% among patients on surveillance. Interestingly a higher mortality rate (14%) was seen in patients receiving chemotherapy alone compared to a 4.6% receiving chemoimmunotherapy.

The most common COVID-19–specific therapies in our dataset were symptomatic treatment with steroids and anticoagulation, tocilizumab (6.5%), remdesivir (2.8%), hydroxychloroquine (3.7%). The use of tocilizumab and remdesivir in our cohort as low compared to other studies [10]. One reason is that our data collection started in the early days of the pandemic when these drugs were not used regularly. Additionally, the availability of the drugs was sparse until the mid of 2020. Remdisivir was given emergency use authorization in May 2020 and gained full FDA approval in October 2020. It only then has this drug become widely available for use.

Morbidity rates from COVID-19 in patients with cancer admitted to the hospital are high, particularly in older patients and those with hematological malignancies. But not all cancer patients are affected equally. These findings allow clinicians to risk-stratify their patients; whether symptomatic or not, and make decisions on social isolation and shielding at appropriate levels. Our data and many other studies demonstrate that patients with hematological malignancies,

**Table 4. Univariate and multivariable Cox-proportional hazard model reporting crude and adjusted ratios (n = 107).**

| Characteristic | Crude HR (95% CI) | p-value | Adjusted HR (95% CI) | p-value |
|---|---|---|---|---|
| Age (years) | | | | |
| >50 | 1 | | 1 | |
| 10–30 | 1.91 (1.84–2.37) | 0.081* | 1.81 (1.40–2.72) | 0.019 |
| 31–50 | 1.17 (1.03–1.56) | | 1.22 (1.14–2.00) | |
| Sex | | | | |
| Female | 1 | 0.107* | - | - |
| Male | 1.84 (1.38–3.05) | | | |
| Active treatment | | | | |
| Yes | 1 | 0.596 | - | - |
| No | 1.19 (0.61–2.33) | | | |
| Treatment | | | | |
| TKI | 1 | | 1 | |
| IV chemotherapy | 1.19 (1.05–2.85) | 0.084* | 1.51 (1.26–2.47) | 0.031 |
| Chemoimmunotherapy | 1.36 (1.10–3.50) | | 1.46 (1.30–2.69) | |
| COVID-19 exposure | | | | |
| No | 1 | | 1 | |
| Contact exposure | 1.49 (1.19–3.17) | 0.058* | 2.18 (1.90–4.44) | 0.001 |
| Travel | 1.77 (1.61–5.12) | | 3.10 (2.73–4.60) | |
| Hospital admission | | | | |
| No | 1 | | | |
| Yes | 1.16 (0.62–2.20) | 0.628 | - | - |
| COVID status at death/discharge | | | | |
| Negative | 1 | | 1 | |
| Not done | 1.80 (1.49–2.38) | 0.024* | 2.28 (1.43–5.63) | 0.001 |
| Positive | 1.88 (1.58–4.16) | | 3.41 (2.12–6.38) | |

particularly acute leukemias, are at high risk for severe COVID-19 infection and mortality. Therefore, preemptive testing, early recognition of infections, and prompt management at a facility with expertise to manage complications are paramount. In addition, protective measures such as universal masks, social distancing, and shielding this susceptible population from COVID-19 exposure are mandatory. Many sites have implemented telemedicine to on-site physical distancing, and these practices should continue if COVID-19 prevalence remains high.

Our study has some limitations. Our analyses are based on patients with hematological malignancy who sought help from centers receiving their treatment. Therefore, this cohort did not capture the outcomes of patients who presented for management at a different hospital. This is particularly true for patients who live in cities or towns and most likely obtained treatment closer to home. Also, we likely missed patients on long-term follow-up who approached their local general physician (GP) or hospital. We, too, were unable to capture asymptomatic patients and found to have COVID-19 positive on screening. In addition, patients on hospice care were not reported or included in this study. Therefore, it is impossible to accurately quantify the burden of infection in patients with hematological malignancy.

Nonetheless, this dataset provides a glimpse of outcomes of patients who presented to a tertiary care hospital in Pakistan where both states of art management for covid infection and the primary malignancy was available. The majority of our patients were on active treatment, and these results help prognosticate; patients who require intensive care carry a very grim prognosis. Data such as ours are especially important in formulating country/region region-specific guidelines regarding the management of covid infections in a specific subset of patients. This

is important in guiding management decisions when resources are limited and medical care is not covered by private insurance.

The expertise in managing COVID-19 infection has evolved over the last year. Early use of dexamethasone, remdesivir, and anticoagulation has improved outcomes.

## Conclusion

In summary, this study of patients with hematological malignancy and COVID-19 accentuates several significant considerations for clinical care and emphasizes the urgent need for more data. Longer-term follow-up and larger sample sizes are needed to understand the effect of SARS-CoV-2 on outcomes in patients with hematological malignancy.

## Supporting information

**S1 Data.**
(XLSX)

## Author Contributions

**Conceptualization:** Adeeba Zaki.

**Formal analysis:** Salman Muhammad Soomar, Danish Hasan Khan.

**Methodology:** Hasan Shaharyar Sheikh, Raheel Iftikhar, Ayaz Mir, Zeba Aziz, Khadija Bano.

**Supervision:** Munira Shabbir-Moosajee.

**Writing – original draft:** Adeeba Zaki, Hasan Shaharyar Sheikh, Raheel Iftikhar, Ayaz Mir, Zeba Aziz, Khadija Bano, Hafsa Naseer, Qamar un–Nisa Chaudhry, Syed Waqas Imam Bokhari.

**Writing – review & editing:** Munira Shabbir-Moosajee.

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
