## [Decision Letter · Decision Letter 0]

27 Dec 2021

PONE-D-21-33433Outcomes of COVID‐19 infection in patients with hematological malignancies- A multicenter analysis from PakistanPLOS ONE

Dear Dr. Zaki,

Thank you for submitting your manuscript to PLOS ONE. After careful consideration, we feel that it has merit but does not fully meet PLOS ONE’s publication criteria as it currently stands. Therefore, we invite you to submit a revised version of the manuscript that addresses the points raised during the review process.

We look forward to receiving your revised manuscript.

Kind regards,

Sinan Kardeş, M.D.

Academic Editor

PLOS ONE

Journal Requirements:

a) Did participants provide their written or verbal informed consent to participate in this study?

“No Funding received”

Reviewers' comments:

Reviewer's Responses to Questions

**Comments to the Author**

1. Is the manuscript technically sound, and do the data support the conclusions?

Reviewer #1: No

Reviewer #2: Partly

Reviewer #3: Partly

Reviewer #4: Partly

Reviewer #5: Partly

Reviewer #6: Partly

2. Has the statistical analysis been performed appropriately and rigorously? 

Reviewer #1: I Don't Know

Reviewer #2: No

Reviewer #3: No

Reviewer #4: No

Reviewer #5: No

Reviewer #6: No

3. Have the authors made all data underlying the findings in their manuscript fully available?

Reviewer #1: Yes

Reviewer #2: Yes

Reviewer #3: Yes

Reviewer #4: Yes

Reviewer #5: Yes

Reviewer #6: Yes

4. Is the manuscript presented in an intelligible fashion and written in standard English?

Reviewer #1: No

Reviewer #2: No

Reviewer #3: No

Reviewer #4: Yes

Reviewer #5: Yes

Reviewer #6: No

5. Review Comments to the Author

Reviewer #1: The authors investigated the outcome of patients with hematological malignancies and Covid-19 infection in a multicenter retrospective cohort in Pakistan. I agree with the authors, that more research in the management of patients with severe immune-suppression and Covid-19 infection is urgently needed.

The collected data in this study are scarce and do not allow any new conclusion.

Unfortunately, this study does not provide any new insights in the field of Covid-19 infection in patients with hematological malignancies.

Reviewer #2: The current study describes 107 patients with COVID-19 and HM retrospectively collected from Pakistan (multicenter). I suggest the following changes to the manuscript:

- Needs extensive English editing.

- What is the rationale for not starting the inclusion criteria from the day of the epidemic start and choosing almost a year after this to start?

- Include the methodology (test details) of the COVID-19 diagnosis.

- The methodology section lacks important details (definitions [disease, variables, outcomes, severity classifications] and detail the statistical analysis.

- Add details of COVID-19 severity and severity classification.

- I suggest including a univariable and a multivariable analysis on the mortality and including important predictors in these analyses.

Reviewer #3: In the manuscript, PONE-D-21-33433, Zaki et al reported the COVID-19 infection outcomes of 107 patients with history of hematological malignancies that were presented at the Oncology Department of 5 tertiary care hospitals in Pakistan from February 2020 to August 2020. The authors summarized the overall clinical outcome, the patient characteristics, clinical presentations, treatments administered, and mortality rate stratified by age, type of malignancy and oncological treatment status for COVID19 in patients with hematological malignancy.

Specific concerns:

1. The authors concluded that their data supports the emerging consensus that patients with hematologic malignancies experience significant morbidity and mortality resulting from COVID-19 infection. However, there is no matched control cases in this study to compare with.

2. No statistical significance (P value) was assessed in the mortality rate analyses.

3. The calculation of mortality rate in Table 2 is wrong, which is the reason why the mortality rate in each group is lower than the overall mortality rate of the study cohort (28%). For example, when stratified for age, the authors said the mortality rate was 7.4%, 5.6% and 10.2% in the age groups 10-30 years, 31-50 years, and 51-70 years respectively. However, the true death rate should be 19.5% (8/41), 17.6 (6/34), and 37.9% (11/29), respectively.

4. Table 1 should be divided into multiple tables.

5. It might be better to present the data in Table 2 by one or more figures.

6. There are sporadic grammar errors throughout the manuscript.

Reviewer #4: Peer-review report

Title: “Outcomes of COVID‐19 infection in patients with hematological malignancies- A multicenter analysis from Pakistan”

Manuscript number: PONE-D-21-33433

Reviewer comment

- Thank you for inviting me to review this paper. I congratulate the authors for writing this research article. Based on their study, the authors describe the outcome of COVID-19 infection in patients with hematological malignancy. However, the document needs revision in the methodology and analysis section before considering the paper for publication.

Abstract

- Please expand the term COVID-19 and SARS-CoV-2

- In the method part, explain how the data is and analyzed and presented

- Replace …‘from February 2020 to August 2020’ with … from February to August 2020’.

- Please spell out numbers when they occur at the beginning of a sentence.

- The authors said that mortality rate is high in patients aged 50 and above without putting any statistical indicator (e.g. p-value). Accordingly, the information seems invalid.

- The conclusion of the study support the knowledge in the existing literature, but it seems relevant to add recommendation. //

Introduction

-explained fairly.

Method

Needs major revision considering the following points

- Clear description of the study setting, study populations, eligibility criteria (inclusion and exclusion criteria), data collection procedure, operational definition (if applicable), and ethical approval of the study (even a little is stated in the declaration section).

- In the statistical analysis part, why the authors preferred to use median over mean? Do the authors check the distribution of the data?

- Similarly, the chi-square test is performed. However, it is unclear where they put the report of this test in the result part.

- With the existing data on your hand, you can consider identifying the factor (s) associated with the COVID-19 outcome (optional).

Result

-Revise based on previous comment on the analysis part. You have to put the chi-square (optional) and the p-value in the stratification of mortality table.

Discussion

-Be consistent on the use of the word COVID-19. You have written as covid 19, COVID-19,...

-Expand the term ASH

- At the end of the discussion part please give a separate sub-heading and label as ‘conclusion’. In this part you can put your conclusion and recommendation (s).

- Revise the lists of abbreviation

Reviewer #5: In this study, Zaki et al, carried out a multicenter, retrospective, cohort study including 107 patients (aged ≥14 years) with diagnosis of hematological malignancy admitted to 5 tertiary care hospitals with laboratory-confirmed and symptomatic COVID-19.

As the authors rightly point out, the value of the study lies in being the first multicenter analysis from Pakistan and aimed at investigating outcome of patients with hematological malignancies and COVID-19 in a context with limited health care resources.

The outcomes in COVID- 19 cancer patients is a topic of great interest, and for this reason, a deeper characterization of the study population is recommended. In particular, the author should specify:

- the duration of follow-up

- the criteria used to define the severity of COVID-19 at admission

- patients with progressive disease, remission or in active therapy

Although in this study, the overall mortality mirrors data from studies carried out in other countries, suggesting a high risk for severe COVID-19 infection and mortality also in these patients, the author should discuss results comparing data with:

- mortality estimates for COVID-19 in the general population, possibly according to sex and stratified for age

- mortality estimates for pre-covid cohort with hematological malignancies, possibly stratified according to sex, age, type of hematological malignancy.

In order to make data analysis clearer to readers, characteristics of the study population should be reported in a table, with information on age/age groups, sex, comorbidities, type of hematological malignancy, disease status and therapy, COVID-19 severity, referred for all patients, survivors and non-survivors. And the results should be discussed accordingly. In particular, the features of the cohort study can be compared according to COVID-19 severity and outcome

Although the sample size makes this study not sufficiently robust to assess events and risk factors that can predict death in this clinical setting, a deeper statistical analysis evaluating association with overall survival of variable such as age, sex, comorbidity, type and status of neoplasia, etc., would help to objectively characterize these patients

Reviewer #6: The article captures data from the real world but has certain limitations, some of which have been highlighted. The comparative mortality and hospitalization statistics in the general population without hematological malignancies should be mentioned for the reader to make a fair comparison. The English should be brushed up. Have edited some of the errors in the attached pdf, but there are more.

6. PLOS authors have the option to publish the peer review history of their article (what does this mean?). If published, this will include your full peer review and any attached files.

Reviewer #1: No

Reviewer #2: No

Reviewer #3: **Yes: **Zhijun Duan

Reviewer #4: No

Reviewer #5: No

Reviewer #6: **Yes: **Prantar Chakrabarti

---

## [Author Response · Author response to Decision Letter 0]

16 Feb 2022

Response to comments

Reviewer #1: The authors investigated the outcome of patients with hematological malignancies and Covid-19 infection in a multicenter retrospective cohort in Pakistan. I agree with the authors, that more research in the management of patients with severe immune-suppression and Covid-19 infection is urgently needed.

The collected data in this study are scarce and do not allow any new conclusion.

Unfortunately, this study does not provide any new insights in the field of Covid-19 infection in patients with hematological malignancies.

Response: This is a first ever multicenter local study conducted in Pakistan, we are presenting the data that we have collected from different settings in Pakistan which represents the local population, and it is highly generalizable to the Cancer patients with COVID-19. The findings may not be new but significant for the country and reflects the patient’s outcomes in an LMIC. 

Reviewer #2: The current study describes 107 patients with COVID-19 and HM retrospectively collected from Pakistan (multicenter). I suggest the following changes to the manuscript:

- Needs extensive English editing.

Response: We have made changes and corrected English errors. See track changes. 

- What is the rationale for not starting the inclusion criteria from the day of the epidemic start and choosing almost a year after this to start?

Response: We have thought of this study since very beginning but as it is a multicenter study it took some time to get the permissions from Departmental and Institutional review boards for data collection and analysis.

- Include the methodology (test details) of the COVID-19 diagnosis.

Response: tested positive for COVID-19 by RT-PCR (page # 6 line # 16)

- The methodology section lacks important details (definitions [disease, variables, outcomes, severity classifications] and detail the statistical analysis.

Response: Added (page # 7 line # 18-23 page # 8 line # 1-25)

Operational definition

Disease

Leukemia is cancer of the body's blood-forming tissues, including the bone marrow and the lymphatic system

Lymphoma is a cancer of the lymphatic system, which is part of the body's germ-fighting network. The lymphatic system includes the lymph nodes (lymph glands), spleen, thymus gland and bone marrow.

Multiple myeloma is a cancer of plasma cells. Normal plasma cells are found in the bone marrow and are an important part of the immune system.

Myelodysplastic syndromes (MDS) are conditions that can occur when the blood-forming cells in the bone marrow become abnormal. This leads to low numbers of one or more types of blood cells.

- Add details of COVID-19 severity and severity classification.

Response:

Clinical classification of suspected or confirmed COVID-19 patients in Pakistan

Asymptomatic SARS CoV-2 infection but with no symptoms. Some asymptomatic patients may be pre-symptomatic if tested early (e.g. as part of contact tracing). 

Non-Severe Oxygen saturation of 94% or greater and respiratory rate of less than 25 breaths/minute. 

Severe Oxygen saturation < 94 % and respiratory rate greater than 25 breath/minute.

Oxygen saturation is maintained by nasal cannulation or simple facemask, no need for non-invasive or invasive mechanical ventilation.

Critical Respiratory compromise severe enough requiring invasive and non- invasive mechanical ventilation. 

Variables

Age, Sex, Hospital, Co-morbidities, Hematological malignancy, treatment, chemotherapy, COVID-19 exposure, presenting complaint, hospital admission, COVID-19 status at death or discharge were the covariates. 

Statistical analysis

The data was entered and analyzed by using STATA version 16. Calculated medians for all continuous variables and frequencies with percentages for categorical variables. Chi-square test was performed to check the association between the covariates and mortality. For survival analysis multivariable Cox-regression hazard regression model was applied and Kaplan Meir survival curve was used to check the overall survival. P-value ≤0.05 was considered significant. 

I suggest including a univariable and a multivariable analysis on the mortality and including important predictors in these analyses.

Response: Univariable and a multivariable analysis on the mortality is done see page # 22 table 4

Reviewer #3: In the manuscript, PONE-D-21-33433, Zaki et al reported the COVID-19 infection outcomes of 107 patients with history of hematological malignancies that were presented at the Oncology Department of 5 tertiary care hospitals in Pakistan from February 2020 to August 2020. The authors summarized the overall clinical outcome, the patient characteristics, clinical presentations, treatments administered, and mortality rate stratified by age, type of malignancy and oncological treatment status for COVID19 in patients with hematological malignancy.

1.The authors concluded that their data supports the emerging consensus that patients with hematologic malignancies experience significant morbidity and mortality resulting from COVID-19 infection. However, there is no matched control cases in this study to compare with.

Response: Thanks for the comment. This is a retrospective observational study where we just have taken the data of patients with hematological malignancy and were COVID-19 positive. We can do matched case control study in future for comparison. 

2. No statistical significance (P value) was assessed in the mortality rate analyses.

Response: Tables are revised. P-value added in the tables by applying chi-square. 

3. The calculation of mortality rate in Table 2 is wrong, which is the reason why the mortality rate in each group is lower than the overall mortality rate of the study cohort (28%). For example, when stratified for age, the authors said the mortality rate was 7.4%, 5.6% and 10.2% in the age groups 10-30 years, 31-50 years, and 51-70 years respectively. However, the true death rate should be 19.5% (8/41), 17.6 (6/34), and 37.9% (11/29), respectively.

Response: The analysis tables are changed to basic tables with chi-square & multivariable survival regression and calculated overall survival.

4. Table 1 should be divided into multiple tables.

Response: Table 1 is changed to baseline characteristics and is stratified on outcome (alive/ dead) and p-values taken out with chi-square. 

5. It might be better to present the data in Table 2 by one or more figures.

Response: The table 2 is changed to treatment and management of patient stratified on outcome (alive/ dead) and p-values taken out with chi-square. 

6. There are sporadic grammar errors throughout the manuscript.

Response: Grammatical errors rectified. 

Reviewer #4: Peer-review report

Title: “Outcomes of COVID‐19 infection in patients with hematological malignancies- A multicenter analysis from Pakistan”

Manuscript number: PONE-D-21-33433

Reviewer comment

- Thank you for inviting me to review this paper. I congratulate the authors for writing this research article. Based on their study, the authors describe the outcome of COVID-19 infection in patients with hematological malignancy. However, the document needs revision in the methodology and analysis section before considering the paper for publication.

Abstract

- Please expand the term COVID-19 and SARS-CoV-2

Response: Explained with detail page #7

- In the method part, explain how the data is and analyzed and presented

Response: The data was entered and analyzed by using STATA version 16. Calculated medians for all skewed continuous variables. Normality of data was check through density plots. Frequencies with percentages for categorical variables. Chi-square test was performed to check the association between the covariates and mortality. For survival analysis multivariable Cox-regression hazard regression model was applied and Kaplan Meir survival curve was used to check the overall survival. P-value ≤0.05 was considered significant. Page # 8.

- Replace …‘from February 2020 to August 2020’ with … from February to August 2020’.

Response: changed from February to August 2020

- Please spell out numbers when they occur at the beginning of a sentence.

Response: Changed. 

- The authors said that mortality rate is high in patients aged 50 and above without putting any statistical indicator (e.g. p-value). Accordingly, the information seems invalid.

Response: The results are changed completely. The baseline details and clinical details are stratified on outcome (alive/dead) and p-value are taken through chi-square. Moreover multivariable analysis was done using Cox-regression. 

- The conclusion of the study support the knowledge in the existing literature, but it seems relevant to add recommendation. //

Introduction

-explained fairly.

Method

Needs major revision considering the following points

- Clear description of the study setting, study populations, eligibility criteria (inclusion and exclusion criteria), data collection procedure, operational definition (if applicable), and ethical approval of the study (even a little is stated in the declaration section).

Response: Data was collected using a standard Performa developed for this study. Demographic, clinical, treatment and laboratory data and serial samples for viral RNA detection were extracted from medical records of 107 patients. Patient with any known hematological malignancy who was positive for COVID-19 on RT-PCR was included in the study. Patients with a clinical or radiological diagnosis of COVID-19, without a positive RT-PCR test were not included in this analysis. Ethical approval was obtained from Ethical Review Committee of the Aga Khan University Hospital, Karachi, Pakistan. Data was retrospectively collected through chart review so there was no need of the informed written consent. 

- In the statistical analysis part, why the authors preferred to use median over mean? Do the authors check the distribution of the data?

Response: Normality of data was check through density plots. For skewed data Median & IQR was calculated. 

- Similarly, the chi-square test is performed. However, it is unclear where they put the report of this test in the result part.

Response: Chi-square test was performed to check the association between the covariates and mortality. P-values reported. 

- With the existing data on your hand, you can consider identifying the factor (s) associated with the COVID-19 outcome (optional).

Response: Thanks for suggestion it can be done in a prospective study. 

Result

-Revise based on previous comment on the analysis part. You have to put the chi-square (optional) and the p-value in the stratification of mortality table.

Response: Chi-square test was performed to check the association between the covariates and mortality. P-values reported.

Discussion

-Be consistent on the use of the word COVID-19. You have written as covid 19, COVID-19,...

Response: changed to COVID-19 in whole manuscript 

-Expand the term ASH

Response: American Society for Hematology (ASH) page #24

- At the end of the discussion part please give a separate sub-heading and label as ‘conclusion’. 

Response: Added a separate section

In this part you can put your conclusion and recommendation (s).

- Revise the lists of abbreviation

Response: Revised abbreviations

Reviewer #5: In this study, Zaki et al, carried out a multicenter, retrospective, cohort study including 107 patients (aged ≥14 years) with diagnosis of hematological malignancy admitted to 5 tertiary care hospitals with laboratory-confirmed and symptomatic COVID-19.

As the authors rightly point out, the value of the study lies in being the first multicenter analysis from Pakistan and aimed at investigating outcome of patients with hematological malignancies and COVID-19 in a context with limited health care resources.

The outcomes in COVID- 19 cancer patients is a topic of great interest, and for this reason, a deeper characterization of the study population is recommended. In particular, the author should specify:

- the duration of follow-up

Response: There was no follow-up as it’s a retrospective study. The suggestion is nice a prospective follow-up study can be done in future. 

- the criteria used to define the severity of COVID-19 at admission

Response: patients with a history of histologically proven hematological malignancies that tested positive for COVID-19 by RT-PCR

- patients with progressive disease, remission or in active therapy

Patients with active therapy/ treatment are included (see table 2). Progressive disease or remission were not recorded. 

Although in this study, the overall mortality mirrors data from studies carried out in other countries, suggesting a high risk for severe COVID-19 infection and mortality also in these patients, the author should discuss results comparing data with:

- mortality estimates for COVID-19 in the general population, possibly according to sex and stratified for age

Response: Mortality recorded in age groups & sex refer to table 1.

- mortality estimates for pre-covid cohort with hematological malignancies, possibly stratified according to sex, age, type of hematological malignancy.

Response: Results are revised characteristics such as age, gender, malignancy & others were check with outcome alive/dead and chi-square test was run to see the significance. 

In order to make data analysis clearer to readers, characteristics of the study population should be reported in a table, with information on age/age groups, sex, comorbidities, type of hematological malignancy, disease status and therapy, COVID-19 severity, referred for all patients, survivors and non-survivors. And the results should be discussed accordingly. In particular, the features of the cohort study can be compared according to COVID-19 severity and outcome

Response: Results are revised characteristics such as age, gender, malignancy & others were check with outcome alive/dead and chi-square test was run to see the significance. Moreover multivariable analysis was performed using Cox-regression to see the survival by regressing all independent variables again outcome alive/dead. 

Although the sample size makes this study not sufficiently robust to assess events and risk factors that can predict death in this clinical setting, a deeper statistical analysis evaluating association with overall survival of variable such as age, sex, comorbidity, type and status of neoplasia, etc., would help to objectively characterize these patients

Response: Multivariable analysis was performed using Cox-regression to see the survival by regressing all independent variables again outcome alive/dead.

Reviewer #6: The article captures data from the real world but has certain limitations, some of which have been highlighted. The comparative mortality and hospitalization statistics in the general population without hematological malignancies should be mentioned for the reader to make a fair comparison. The English should be brushed up. Have edited some of the errors in the attached pdf, but there are more.

Response: Grammatical editing has been done.

---

## [Decision Letter · Decision Letter 1]

9 Mar 2022

PONE-D-21-33433R1Outcomes of COVID‐19 infection in patients with hematological malignancies- A multicenter analysis from PakistanPLOS ONE

Dear Dr. Zaki,

Thank you for submitting your manuscript to PLOS ONE. After careful consideration, we feel that it has merit but does not fully meet PLOS ONE’s publication criteria as it currently stands. Therefore, we invite you to submit a revised version of the manuscript that addresses the points raised during the review process.

We look forward to receiving your revised manuscript.

Kind regards,

Sinan Kardeş, M.D.

Academic Editor

PLOS ONE

Journal Requirements:

Reviewers' comments:

Reviewer's Responses to Questions

**Comments to the Author**

1. If the authors have adequately addressed your comments raised in a previous round of review and you feel that this manuscript is now acceptable for publication, you may indicate that here to bypass the “Comments to the Author” section, enter your conflict of interest statement in the “Confidential to Editor” section, and submit your "Accept" recommendation.

Reviewer #1: (No Response)

Reviewer #3: All comments have been addressed

Reviewer #4: All comments have been addressed

Reviewer #5: All comments have been addressed

2. Is the manuscript technically sound, and do the data support the conclusions?

Reviewer #1: Yes

Reviewer #3: Yes

Reviewer #4: Yes

Reviewer #5: Partly

3. Has the statistical analysis been performed appropriately and rigorously? 

Reviewer #1: Yes

Reviewer #3: Yes

Reviewer #4: Yes

Reviewer #5: Yes

4. Have the authors made all data underlying the findings in their manuscript fully available?

Reviewer #1: Yes

Reviewer #3: Yes

Reviewer #4: Yes

Reviewer #5: Yes

5. Is the manuscript presented in an intelligible fashion and written in standard English?

Reviewer #1: No

Reviewer #3: No

Reviewer #4: No

Reviewer #5: No

6. Review Comments to the Author

Reviewer #1: The authors of the manuscript "Outcomes of COVID‐19 infection in patients with hematological malignancies- A multicenter analysis from Pakistan“ have partially addressed previous comments. The manuscript still needs intensive language editing.

Reviewer #3: The authors have addressed most of the concerns from the 1st-round review, and this reviewer is satisfied with the improved quality of the revised manuscript. However, the English language skill still needs to be improved.

Reviewer #4: Dear /sir, most of the comments given were addressed correctly. I have a few suggestions to improve the readability of this manuscript before publication is commenced. My concerns are:-

- Please expand the term COVID-19 and SARS-CoV-2 at the beginning of the abstract page 3 line 3

- Replace …‘from February 2020 to August 2020’ with … from February to August 2020’ page 3 line 11.

- The information included in the operational definition must be cited with appropriate literature/s.

- Authors should check for typos and grammatical errors throughout the manuscript which need to be edited after careful reading.

Reviewer #5: I sugest the authors to keep the first 3 paragraph of Baseline characteristics, taking care to integrate the information added in the revised version to avoid repetitions

The comparative mortality and hospitalization statistics in the general population should be mentioned for the reader to

make a fair comparison.

English should be improved

7. PLOS authors have the option to publish the peer review history of their article (what does this mean?). If published, this will include your full peer review and any attached files.

Reviewer #1: No

Reviewer #3: No

Reviewer #4: No

Reviewer #5: No

---

## [Author Response · Author response to Decision Letter 1]

17 Mar 2022

Reviewer #1: The authors of the manuscript "Outcomes of COVID‐19 infection in patients with hematological malignancies- A multicenter analysis from Pakistan“ have partially addressed previous comments. The manuscript still needs intensive language editing.

Response: Grammatical corrections are made, see track changes 

Reviewer #3: The authors have addressed most of the concerns from the 1st-round review, and this reviewer is satisfied with the improved quality of the revised manuscript. However, the English language skill still needs to be improved.

Response: Grammatical corrections are made, see track changes

Reviewer #4: Dear /sir, most of the comments given were addressed correctly. I have a few suggestions to improve the readability of this manuscript before publication is commenced. My concerns are:-

- Please expand the term COVID-19 and SARS-CoV-2 at the beginning of the abstract page 3 line 3

Response: Changed to Severe acute respiratory syndrome coronavirus 2 (SARS-CoV-2). Coronavirus disease of 2019 (COVID-19) page #3 line #3-4

- Replace …‘from February 2020 to August 2020’ with … from February to August 2020’ page 3 line 11.

Response: Changed to February to August 2020

- The information included in the operational definition must be cited with appropriate literature/s.

Response: Cited all operational definitions

- Authors should check for typos and grammatical errors throughout the manuscript which need to be edited after careful reading.

Response: Grammatical corrections are made, see track changes

Reviewer #5: I suggest the authors to keep the first 3 paragraph of Baseline characteristics, taking care to integrate the information added in the revised version to avoid repetitions.

Response: Made two sections added first 3 paragraphs with their tables in baseline characteristics and rest in multivariable analysis. 

The comparative mortality and hospitalization statistics in the general population should be mentioned for the reader to make a fair comparison.

Response: However, in the general population it range from 9-29.6%. In white Hispanic group the morality was less than 10% while in Black and Hispanic groups it was less than 30% in the general population. Page 18 line# 6-8

---

## [Decision Letter · Decision Letter 2]

4 Apr 2022

Outcomes of COVID‐19 infection in patients with hematological malignancies- A multicenter analysis from Pakistan

PONE-D-21-33433R2

Dear Dr. Zaki,

We’re pleased to inform you that your manuscript has been judged scientifically suitable for publication and will be formally accepted for publication once it meets all outstanding technical requirements.

Kind regards,

Sinan Kardeş, M.D.

Academic Editor

PLOS ONE

Additional Editor Comments (optional):

Reviewers' comments:

Reviewer's Responses to Questions

**Comments to the Author**

1. If the authors have adequately addressed your comments raised in a previous round of review and you feel that this manuscript is now acceptable for publication, you may indicate that here to bypass the “Comments to the Author” section, enter your conflict of interest statement in the “Confidential to Editor” section, and submit your "Accept" recommendation.

Reviewer #3: All comments have been addressed

Reviewer #4: All comments have been addressed

Reviewer #5: All comments have been addressed

2. Is the manuscript technically sound, and do the data support the conclusions?

Reviewer #3: Yes

Reviewer #4: Yes

Reviewer #5: Yes

3. Has the statistical analysis been performed appropriately and rigorously? 

Reviewer #3: Yes

Reviewer #4: Yes

Reviewer #5: N/A

4. Have the authors made all data underlying the findings in their manuscript fully available?

Reviewer #3: Yes

Reviewer #4: Yes

Reviewer #5: Yes

5. Is the manuscript presented in an intelligible fashion and written in standard English?

Reviewer #3: Yes

Reviewer #4: Yes

Reviewer #5: Yes

6. Review Comments to the Author

Reviewer #3: (No Response)

Reviewer #4: (No Response)

Reviewer #5: The authors of the manuscript "Outcomes of COVID‐19 infection in patients with hematological malignancies- A multicenter analysis from Pakistan“ have addressed previous comments.

7. PLOS authors have the option to publish the peer review history of their article (what does this mean?). If published, this will include your full peer review and any attached files.

Reviewer #3: No

Reviewer #4: No

Reviewer #5: No

---

## [Editor Report · Acceptance letter]

6 Apr 2022

PONE-D-21-33433R2 

“Outcomes of COVID‐19 infection in patients with hematological malignancies- A multicenter analysis from Pakistan” 

Dear Dr. Zaki:

I'm pleased to inform you that your manuscript has been deemed suitable for publication in PLOS ONE. Congratulations! Your manuscript is now with our production department. 

Kind regards, 

on behalf of

Dr. Sinan Kardeş 

Academic Editor

PLOS ONE